# Household-Reported Availability of Drinking Water in Africa: A Systematic Review

**Mair L. H. Thomas** [1,*] **, Andrew A. Channon** [2] **, Robert E. S. Bain** [3] **, Mutono Nyamai** [4] **and Jim A. Wright** [1]

1   Geography and Environmental Science, University of Southampton, Shackleton Building 44, Southampton SO17 1BJ, UK; J.A.Wright@soton.ac.uk
2   Social Statistics and Demography, University of Southampton, Murray Building 58, Southampton SO17 1BJ, UK; A.R.Channon@soton.ac.uk
3   Division of Data, Analysis, Planning and Monitoring, UNICEF, 3 United Nations Plaza, New York, NY 10017, USA; rbain@unicef.org
4   The Wangari Maathai Institute for Peace and Environmental Studies, University of Nairobi, Upper Kabete, Kapenguria Road, Nairobi P.O. Box 30197, Kenya; mutono.nyamai@wsu.edu
*   Correspondence: M.L.H.Thomas@soton.ac.uk

**Abstract:** Domestic drinking water supplies prone to interruptions and low per capita domestic water availability have been frequently reported among African households. Despite expanded international monitoring indicators that now include metrics of water availability, the range of methods used for measuring and monitoring availability remains unclear in Africa. Few household surveys have historically assessed water continuity and per capita availability, and both pose measurement challenges. This paper aims to examine the methods used to measure availability and synthesise evidence on African domestic water availability by systematically reviewing the literature from 2000–2019. Structured searches were conducted in five databases: Web of Science Core Collection, Scopus, GEOBASE, Compendex and PubMed/Medline. A total of 47 of 2406 reports met all inclusion criteria. Included studies were based on empirical research which reported the household's perspective on a water availability measure. Most studies had methodological problems such as small sample sizes, non-representative sampling and incomplete reporting of methods and measures of uncertainty. Measurement of drinking water availability is primarily reliant on quantifying litres/capita/day (LPCD). Only four (9%) of the included studies reported an average water availability over the international benchmark of 50 LPCD. This pattern of water insufficiency is broadly consistent with previous studies of domestic water availability in Africa. The review highlights the need for high-quality and representative studies to better understand the uncertainties and differences in household water availability across Africa, and the methods used to measure it.

**Keywords:** drinking water; water supply; availability; sufficiency; continuity; intermittency; litres per capita per day; African Union

## 1. Introduction

For billions of people worldwide, the poor availability of drinking water restricts consumption patterns and affects quality of life. At present, globally, at least one billion people experience an interruption to their supply throughout a 24 h period [1] and around 3.1 billion individuals depend on unreliable, non-piped water supplies that are located off-premises [2]. Despite international progress towards achieving universal access to drinking water, in sub-Saharan Africa (SSA), only 57% of the population report having an improved water supply that is fully functional, available when needed,

easily accessible and provides good quality and safe water [3]. Currently there is insufficient data in Africa to estimate the population using a water source that is "available when needed", suggesting a need for evidence to fill this gap [4]. In addition, systematic evidence about how related measurement criteria are used in studies of water availability is limited, as is the extent to which reported availability varies by setting or study design. This systematic review aims to (i) assess drinking water availability across Africa from a household's perspective; and (ii) examine methods used to measure drinking water availability.

The World Health Organization (WHO) outline a global drinking water availability benchmark which recommends that between 50 and 100 L/capita/day (LPCD) is required to meet domestic needs, including washing, personal hygiene and cleaning [5]. Meeting this benchmark is crucial, especially given an estimated 368,000 deaths annually in SSA are attributed to water, sanitation and hygiene (WASH) related diarrhoea [6]. Women and children are often most affected by poor water availability due to cultural and traditional expectations, such as the burden of collecting water [7,8]. Further potential implications include poor educational attainment [5], loss of employment opportunities and subsequent productivity and economic losses to households and society [9].

Quantifying and measuring availability is complex, not least because in many African and low-income settings many households use multiple water sources, which in turn, creates multiple sites of measurement. Methods for measuring non-metered supplies are not standardised, with recall periods for questions on water usage ranging from 24 h to a week, and others using consecutive daily visits to monitor water consumption or direct observation [2]. In some instances, households have been asked to quantify their water by placing stones in buckets every time a water container of a known size is filled [10], or to use picture prompts of local water containers of known sizes for quantification [11]. Calls have been made [2] for the 'end goal' of measuring water use to be quantification using LPCD, which has been shown to be used in multiple contexts but by no means across the board.

The acceptance of availability as a key concept of WASH is also reflected in the sixth Sustainable Development Goal (SDG), which is closely linked to the Human Right to Water [12], and aims to ensure availability and sustainable management of water and sanitation supplies for all, by 2030. Critically, unlike its predecessor, target 7.C of the Millennium Development Goals (MDG), it considers availability. The recent incorporation of drinking water availability into international monitoring indicators has been facilitated through the inclusion of core questions, developed by the Joint Monitoring Programme (JMP) of the WHO and United Nations Children's Fund (UNICEF) [13], within major international survey programmes. These are designed to provide global measurement standards to ensure consistent monitoring through the standardisation of information collected within household surveys and censuses. Core questions are supplemented by expanded questions that can be asked should resources allow, and response categories have been designed to be universally applicable. The JMP's core question which addresses water availability, asks 'In the last month, has there been any time when your household did not have sufficient quantities of water when needed?' [14]. Expanded questions focus on aspects of unavailability, continuity, intermittency and household water tanks, as well as seasonal variations in availability (see Appendix A, Table A1).

As the diversity of core and expanded questions illustrates, service availability is increasingly seen as a critical WASH element. However, while the JMP has thus far been relatively successful in streamlining their core and expanded questions into international survey programmes, a greater understanding of the suitability of the methods used to measure availability in Africa, in light of the JMPs new set of questions, is merited. Given the importance of capturing the viewpoint of the user, examination of the full spectrum of approaches used to measure components of water availability is also warranted. Given this, this systematic review will help in evaluating the effectiveness of the JMP's core questions and most importantly, provide a synthesis of the existing evidence on availability of drinking water supplies in the African context.

Several related systematic reviews explore the reliability of supplies [15], deficiencies in supply systems [16] and methodologies used to measure water availability [2]. However, two of these review

the implications of poor water availability for health [2,16], whilst the third examines population coping strategies in response to insufficient supplies [15]. Similarly, the scope is either global [16] or primarily focuses on low and middle income countries (LMICs) [2,15], rather than regional. Often related reviews exclude certain water supply types, or focus solely on domestic piped supplies [16]. No known review has compared availability in the context of improved and unimproved supplies. While Tamason et al. [2] explored the methods used to measure household water availability, the review predates revised core questions and results are not assessed against the WHO's global availability benchmark. Consequently, this review aims to address the following research questions:

- How does household-reported water supply availability vary between urban, peri-urban and rural areas of Africa?
- Are households across Africa meeting the WHO benchmark for water availability?
- Does household-reported water availability and the prevalence of interruptions differ for improved versus unimproved water source types?
- What methods are currently being used to measure household water availability as defined by the JMP?

## 2. Materials and Methods

### 2.1. Literature Search Methods

The systematic review was undertaken following PRISMA guidelines [17]. A protocol is available via the PROSPERO register of systematic reviews (ID# 124139). During February 2019 a search of electronic databases, including Web of Science Core Collection, Scopus, GEOBASE, Compendex, PubMed/Medline, was undertaken by one reviewer (MT). In addition, backward citation tracking was undertaken for all included studies. Search terms related to water availability, drinking water, interruptions, types of water source and households (Appendix B; Box A1) and listed African Union countries (Appendix B; Box A2). Searches used the following structure:

[Continuity/interruptions/availability] AND [domestic water] AND [water supply type]
AND [African Union country]

Thus, terms for each of these four concepts, including truncation terms, for example 'contin*' and 'reliab*'were combined with OR, before being combined with AND.

### 2.2. Inclusion and Exclusion Criteria

Any study reporting an improved or unimproved groundwater or piped drinking water source was included. Studies reporting only packaged water, rainwater or surface water (such as rivers, dams, lakes or ponds) were excluded due to such sources' frequent use as alternatives in the absence of other improved source types [18]. Only studies undertaken in the 55 African Union countries were included [19], as were those in English, Portuguese and French languages.

Studies were included that reported any of three pre-defined measures of drinking water availability: (1) quantities of water available and/or used within a given period, (2) hours of supply a day and (3) frequency of supply breakdowns. These measures were developed in alignment with the WHO's global benchmark and are based on the literature, as well as both the JMP's core and expanded questions.

The review's primary focus was on the experiences and perspective of domestic users of water supplies; therefore, its focus was only on domestic water availability reported by households or individuals. Studies of water availability in schools, healthcare facilities and agricultural or industrial water supply were excluded. Similarly, domestic water supply interruptions that were reported by service providers, rather than households or individuals, were excluded.

This review focused on academic research studies and the following research report types were included; peer-reviewed journals, conference proceedings and theses. Conference proceedings were

included to try to reduce the effect of publication bias [20]. Studies based on data collected from 2000 onwards were included; this marks the year that the JMP developed the standard set of drinking water assessment criteria [21] and when international monitoring of WASH targets began through the MDGs. All forms of study design and research methods were included, for example, focus groups, questionnaires, surveys, interviews.

### 2.3. Study Selection

All titles and abstracts of the retrieved citations were exported by MT into Endnote X8.2 and duplicates were identified and removed. Following initial screening of abstracts and titles, full texts were screened and characterised. For all studies in English, screening was undertaken by MT using the Metagear package [22] in the software R. JW screened six studies in French, no Portuguese studies were returned in the searches.

Following preliminary screening, a random sample of 10% ($n$ = 240) of all studies were independently screened by one of MN, AC and JW (henceforth referred to as the secondary reviewers). The results from the secondary reviewers' screening and MT's screening were cross-checked to identify any discrepancies. Abstracts and titles of reports with such discrepancies were re-reviewed by MT and a revised inclusion decision made based on only their titles, keywords and abstracts. Cohen's Kappa (k) was calculated to determine the level of agreement between MT's and the secondary reviewers' screening of the sub-sample of reports.

Full-text articles were screened for all studies included on the basis of abstract and title. Where any uncertainty by MT regarding a study's inclusion occurred, the studies were independently reviewed by both JW and AC. A final decision regarding inclusion was then made following a joint discussion.

### 2.4. Data Extraction

At the characterisation stage, five different categories of information were collected and recorded from all eligible studies. Basic descriptive information, such as title, author, year of publication and research objectives, were extracted. Studies were also classified based on their study design; the sample size, sampling strategy and design of each study were all collated. Any form of random sampling (e.g., simple random, systematic, and stratified-random) was classified as representative. All other sampling strategies (i.e., convenience, purposive and quota sampling) were classified as non-representative. Characteristics relating to the study's research setting and population were also extracted, for example, country, location, and the size and type of settlement.

The water supply type and supplier (e.g., private, public, community, self, non-governmental organisation (NGO)) were also collected. The water supply used by the majority of the study population in each study was recorded as the 'dominant water source'. Each dominant water supply type was reported based on the JMP's improved/unimproved classification. Where a study reported multiple water supplies, these were grouped and classified as either improved or unimproved. Required conversion factors, such as for converting numbers of jerry cans of water into litres were drawn from published estimates [23].

Data were extracted to evaluate which of the three measures of water supply availability were used. Intermittent supplies were differentiated from supply breakdowns by the study's description of a supply's availability. For example, study reports using terms such as 'broken' or 'breakdown' were characterised as breakdowns. Where reported, the question(s) used by the study was extracted (see: Table 1), together with the recall period and nature of the household respondent. Measures of uncertainty of estimates, such as standard error and confidence intervals, were also collected, together with household size.

For all included qualitative studies, data extraction entailed the collection of first order constructs (participants' quotes) and second order constructs, which are researcher 'interpretations, statements, assumptions and ideas' based on first order constructs [24]. Where the distinction between the two types of construct was not evident, then information was classified as a second order construct [25].

**Table 1.** Measures of water availability.

| | Measure of Water Availability | Questions Used by the Included Study | Example Reported Metric of Availability Used in the Study |
|---|---|---|---|
| **1** | Quantity of water used by households | Sufficient water available during last month? | Yes/No |
| | | Do you have a sufficient amount of water to cover your needs? | Yes/No |
| | | On average, how much water do you use in a day? | Litres per capita per day |
| | | | % of households > XX litres per capita |
| **2** | Average hours of service | On average, for how many hours a day/days a week is your supply available? | XX hours per day |
| | | | % of households > 12 h per day (or >4 days a week?) |
| | | Water available "continuously" | Yes/No |
| **3** | Frequency of breakdowns | Is water available from the main water supply at the time of the study? | Yes/No |
| | | Main supply functioning? | % of households reporting yes |
| | | Can you predict when an interruption is going to happen to your water supply? | Yes/No |

## 2.5. Critical Appraisal of Study Quality

Study quality was assessed based on ten criteria in the STROBE Statement [26] (see Appendix C, Table A2), with an additional four criteria for quantitative studies only. No study was excluded based on it being low quality. Instead, the influence of study quality issues, such as non-representative sampling, were considered through narrative synthesis.

Due to reporting complexity and diversity in the included studies, formal tests for publication bias, such as funnel plots, and for heterogeneity, such as Higgins $I^2$, could not be undertaken. For example, most studies did not report measures of confidence or standard errors for summary statistics. As a result, representativeness of the sampling strategy and sample size were used as indicators of study precision and error.

As an additional study quality measure, where studies failed to report household sizes, but did report both litres/household/day (LPHHD) and LPCD, the household size was calculated and its plausibility assessed based on contextual knowledge of the study location. For example, a seemingly implausible household size of 30 for a study in northern Ghana [27] could have reflected homesteads consisting of an extended household structure, linked through kinship to the same head of household.

## 2.6. Water Scarcity Variable

In preparation for the quantitative analysis, which was undertaken on the results of the included studies, a water scarcity variable was created. Names of study site locations were geocoded with town/city or neighbourhood precision using the MapQuests Application Programming Interface (API) [28]. Coordinates were cross-checked against reported study site names and any geocodes less precise than neighbourhood level were manually reviewed and geocoded. The resultant coordinates were linked, using ArcMap 10.7, to the Water Footprint Network's (WFN) blue water scarcity map layer [29], which comprises a ratio of blue water consumption to blue water availability. This data was used as it is publicly available, has undergone peer-review and most closely relates to the implementation years of included studies. Water scarcity was approximated for each of the geocoded locations, using a buffer based on the area of the study site. Where a study had multiple study sites (*n* = 9), an average water scarcity score was calculated.

*2.7. Analysis*

Given the diversity of included studies, a balance between narrative synthesis, thematic discussion and explorative quantitative analysis was undertaken [30]. For the first measure of availability (LPCD and LPHHD) elements of a meta-summary have been undertaken. Qualitative findings have been quantitatively aggregated by extracting, grouping and formatting findings that related to the standard of the water supply and locality. In contrast, analysis of the second two measures of availability (hours of supply a day and frequency of supply breakdowns) provides a more narrative understanding of the experiences and perceptions of water availability, especially with regards to interruptions and continuity of supply.

Insufficient data and a lack of reporting of measures of uncertainty (i.e., standard deviations for estimates or confidence intervals for summary means and proportions) precluded the use of quantitative meta-analysis techniques such as meta-regressions and forest plots. In the absence of uncertainty measures, mean water consumption (LPCD) per study was plotted on a bubble chart to visualise the study findings analogously to a forest plot. Where studies reported multiple figures for different population sub-groups rather than mean consumption in LPCD, a population-weighted average was calculated.

Since LPCD was the most commonly reported availability metric, single country studies reporting LPCD were included in a multiple linear regression analysis. This was undertaken in Stata version 16.1 and predicted LPCD from locality (i.e., rural/urban), gross domestic product (GDP) per capita, purchasing power parity (PPP) in constant 2017 prices (USD), if the study was an intervention, water supply type (i.e., piped/other) and water scarcity. Prior to running the multiple linear regression model, exploratory bivariate regressions were undertaken between LPCD and each of the explanatory variables.

In the multiple linear regression, freshwater scarcity was included since it may have restricted drinking water availability, especially in Africa where seasonal variations and drought are common [31]. Piped supply was included since more water may be consumed when piped to the yard or home rather than fetched from further away [32]. GDP per capita (PPP) [33] for each study's publication year was used as a proxy for development level [34], which affects national WASH investment [35]. Rurality of study site was included since rural areas typically have lower WASH infrastructure levels and, thus, lower water availability [3]. Studies evaluating interventions were also examined through a binary covariate, since such interventions often increase water availability and consumption. Standard regression assumptions were checked, and the model was re-estimated to exclude any outliers that had a large influence on the coefficient estimates.

## 3. Results

A total of 4948 reports were returned from the database searches (Figure 1). In addition to this, 13 other reports were identified from backwards reference searching, resulting in 4961 studies for review. Of these, 2555 were duplicates.

Following the removal of duplicates, 2406 reports were screened by title and abstract, of which 68 reports met all criteria, whilst 74 were classified as being ambiguous and needed further analysis before a definitive decision could be made. A total of 2264 reports were excluded at this initial screening stage. Of these, 66% did not meet any of the inclusion criteria, 5% (*n* = 113) did not relate to African Union countries, 8% (*n* = 186) focused on water quality and 9% (*n* = 198) reported on rainwater, packaged water or unimproved sources (Appendix D; Figure A1). Additional reasons for reports being excluded were due to their focus on the accessibility of drinking water supplies, sanitation, or health-related implications of water. Some studies did not report the household perspective but gave that of the utility company or supplier.

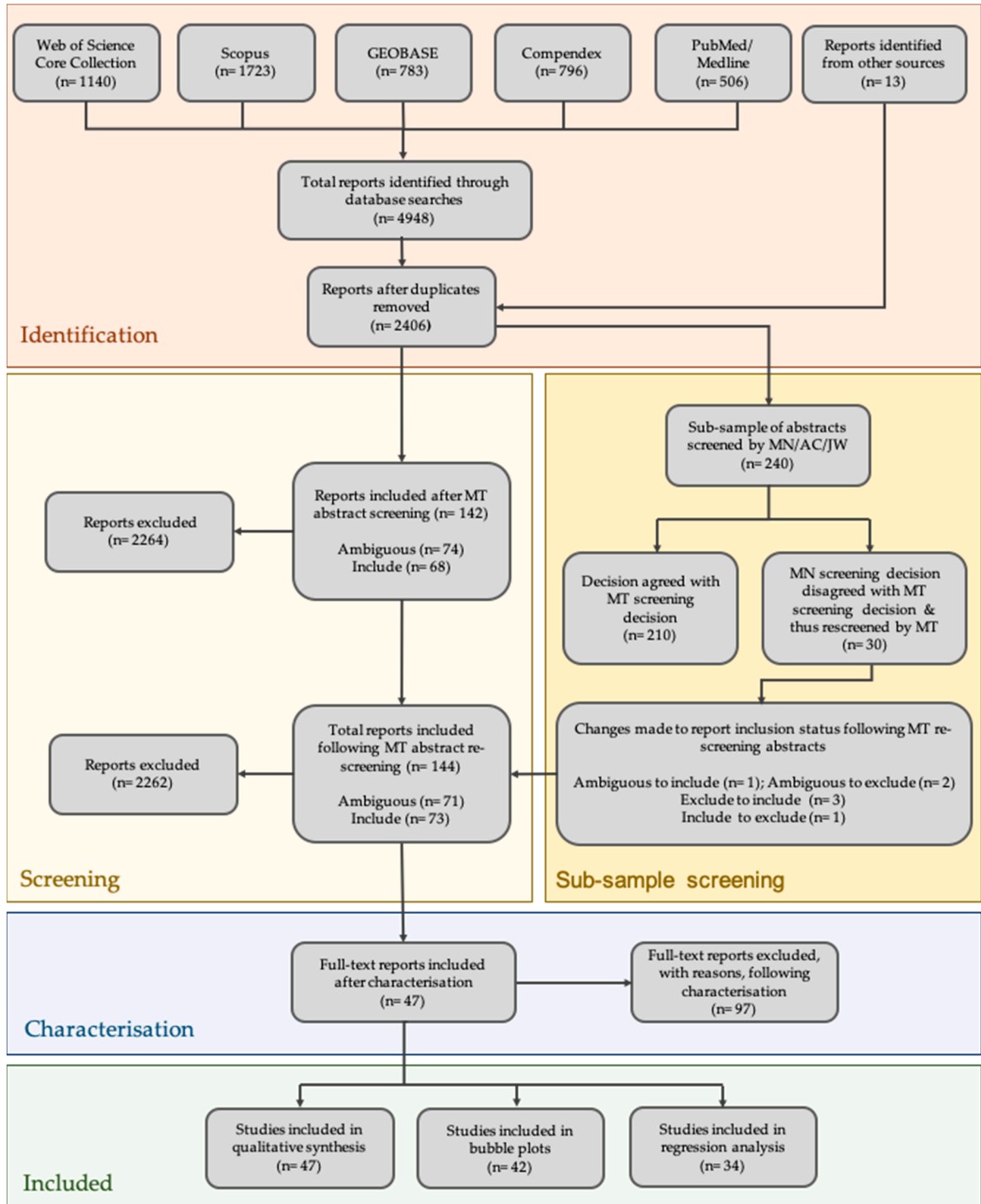

**Figure 1.** Flowchart of study selection process.

Of the sub-sample of reports (*n* = 240) screened by secondary reviewers, there were thirty reports for which the secondary reviewer's decision differed from MT's initial screening decision (Table 2). The secondary reviewers disagreed with the inclusion of 33% of reports MT included based on their titles and abstracts. They felt that 11% of the reports MT excluded should be included. A Kappa statistic suggested fair agreement between MT and the secondary reviewers' judgment regarding the inclusion of these reports, k = 0.376 (*p* = < 0.005). For the purpose of calculating this test statistic, three reports classified as ambiguous were excluded, therefore *n* = 237.

**Table 2.** Cross-tabulation of MT's screening results and the results from MN/JW/AC's screening of a 10% sub-sample.

| | | MT Screening Results | | | |
|---|---|---|---|---|---|
| | | Include | Exclude | Ambiguous | Total |
| MN/JW/AC screening results | Include | 10 | 24 | 0 | 34 |
| | Exclude | 3 | 200 | 3 | 206 |
| | Total | 13 | 224 | 3 | 240 |

Note: Shaded figures represent those which MT and MN/JW/AC (*n* = 30) disagreed on following their independent screening of a 10% sample of reports.

Reanalysis of the thirty papers disagreed upon by MT and the secondary reviewers resulted in one report originally classed as ambiguous being included due to its mention of availability, and two reports which were originally ambiguous being excluded. These reports were excluded because one focused on water quality and its safety and the other modelled willingness to pay for water. Three out of the 24 reports were reincluded after MT originally excluded them. Human error was the cause of these reports having been excluded during MT's original abstract screening stage. One report was excluded which MT had initially included. Therefore, 144 reports were included for subsequent characterisation.

Analysis of the full texts resulted in the inclusion of 47 studies which met all inclusion criteria. A further 97 reports were excluded (Appendix D; Figure A2); 22% did not meet all the inclusion criteria. 32% discussed water availability, however they did not report one of the pre-defined availability measures and thus were excluded. A further 14% did not give the household perspective, while 11% focused on accessibility to water. Thirty-four studies reporting LPCD were included in regression analysis after excluding one such multi-country study.

### 3.1. Study Quality

Among the fourteen study quality criteria (Table 3), there were some generic strengths, such as the study rationale being stated, and generic weaknesses, for example reporting of precision. More than three quarters (81%) reported the sampling strategy used, but methods were rarely described in enough detail to enable replication.

Of criteria specific to quantitative studies, all studies failed to account for missing data, 91% did not describe statistical methods used therefore preventing replication and 97% did not justify their choice of variables. Furthermore, most studies did not report levels of precision, preventing meta-analysis or forest plot production.

**Table 3.** Assessment of study quality.

| | Item | Criterion | Content | Met Criteria (% of Total Studies) |
|---|---|---|---|---|
| Core criteria for all studies (*n* = 47) | 1 | Rationale for study stated | Study objective clearly stated | 100% |
| | 2 | Rationale for chosen participants | Eligibility criteria for study participants described. | 64% |
| | 3 | Water supply characteristics documented | Water source/supply characteristics used by study participants documented. | 94% |
| | 4 | Sampling strategy reported | Sampling strategy described (e.g., purposive; simple random; multi-stage etc.). For secondary data studies; sources and dates of access are given. | 81% |
| | 5 | Arithmetic error has been eliminated | All descriptive statistics and percentages reported were arithmetically correct. | 89% |
| | 6 | Limitations and bias | Limitations of the study are discussed, taking into account potential bias and precision. | 23% |
| | 7 | Participants recorded throughout | Numbers of individuals at each study stage reported (e.g., those ineligible; declining to participate; unavailable for interview). | 17% |
| | 8 | Justification for methods | Choice of interviews, questionnaires, focus groups, surveys etc. justified. | 32% |
| | 9 | Replicability | Methods suitably described to enable replication. | 23% |
| | 10 | Extraction of data and themes | Conclusions have been explained and justified. | 96% |
| Criteria for quantitative studies only (*n* = 42) | 11 | Statistical methods | All statistical methods described in sufficient detail to enable replication. | 9% |
| | 12 | Justification for variables | Use of all chosen quantitative variables explained. | 3% |
| | 13 | Missing data | Missing data acknowledged, and how addressed is clearly explained. | 0% |
| | 14 | Precision | Confidence intervals, odds ratios, standard errors and/or standard deviations reported for water continuity/availability estimates. | 12% |

### 3.2. Study Characteristics

Most studies were published from 2010 onwards (66%), with the majority being post-2015 (41%), and most were cross-sectional (87%) (Table 4). Quantitative household surveys were the dominant research method used to collect information on water availability (43%). In some cases [36], additional methods such as unspecified interviews (meaning it was unknown whether they were structured or unstructured) were used to complement the household survey and gain greater detail on water availability. Structured questionnaires were also frequently used (15%); Katsi et al. [37] used complementary participatory rural appraisal techniques to verify results. A tenth of the studies (11%) used unspecified interviews and a further 11% used unspecified questionnaire designs. Water-meters were used by three studies, all of which examined piped water supplies.

Studies reported sample sizes as the number of households and/or the number of individuals (Appendix E, Table A3). The majority (77%) reported samples of households (min. = 15, max. = 20,000, S.D. = 3302.7)). A total of 11% of studies reported both individual and household sample sizes. Excluding Adekalu et al.'s [38] sample of 20,000 households, the mean sample size was 316 (min. = 20, max. = 1860, S.D. = 406.7). In comparison, 23% of studies reported samples of individuals; the mean sample size was 431 people (min. = 4, max. = 1080, S.D. = 364.3). Half of the studies (55%) used non-representative sampling techniques such as convenience sampling (6%) (min. = 40, max. = 653, S.D. = 289.7) and purposive sampling (19%) (min. = 20, max. = 683, S.D. = 198.3). In addition, multi-stage sampling was used by 26% of studies (min. = 114, max. = 1203, S.D. = 337.9), whereas Booysen et al. [39] reported using quota sampling. Representative sampling methods were undertaken by 36% of studies, and included methods such as systematic sampling (min. = 50, max. = 1015, S.D. = 538.9), stratified random sampling (min. = 674, max. = 1860, S.D. = 838.6) and simple random sampling (min. = 4, max. = 1080, S.D. = 371.3). Examples of systematic random sampling methods included choosing every tenth house along a street [40], or every fifth household when starting from a junction [41]. Four studies failed to report the sampling strategy used.

A total of 61% of studies reported the dominant water supply as being improved. Piped water connections were reported by 39% of studies (the precise location of this supply was often unstated, i.e., whether or not it was from a neighbour, in the yard or within the dwelling), while other dominant improved supplies included boreholes (15%), dug wells (6%) and public standpipes (6%). Unimproved or ambiguously defined water supplies primarily included unclassified wells (21%). Examples of unclassified wells include Malian wells [42] and marigots [43]. Vended water supplies were reported by 4% of studies and included communal water kiosks and tankers.

Figure 2 shows most studies were undertaken in eastern and western Africa, with 17% of studies in Nigeria. In eastern Africa, Kenya and Tanzania were the most frequently studied countries. In southern Africa, the studies were undertaken in South Africa (9%), Eswatini (2%), Zimbabwe (9%) and Botswana (4%). The only study country in northern Africa was Algeria where 9% of studies were undertaken, likewise in central Africa, Cameroon was the only study country. Half of the studies (51%) explored drinking water availability in urban areas, 40% in rural areas and 9% in peri-urban areas. No study specifically considered informal settlements.

**Table 4.** Characteristics of included studies.

| | Study Characteristics | % of Included Studies (*n*) |
|---|---|---|
| **Basic characteristics** | *Year of publication* | |
| | 2000–2004 | 10.6% (5) |
| | 2005–2009 | 23.4% (11) |
| | 2010–2014 | 25.5% (12) |
| | 2015–2019 | 40.5% (19) |
| | *Study design* | |
| | Case-control | 2.1% (1) |
| | Cohort study | 2.1% (1) |
| | Cross-sectional | 87.2% (41) |
| | Longitudinal | 6.4% (3) |
| | Quasi-experimental | 2.1% (1) |
| | *Sampling* | |
| | Representative | 36.2% (17) |
| | Non-representative | 55.3% (26) |
| | Not stated | 8.5% (4) |
| | *Dominant method* | |
| | Household survey | 42.6% (20) |
| | Household water-meters | 6.4% (3) |
| | Semi-structured interview | 4.3% (2) |
| | Semi-structured questionnaire | 6.4% (3) |
| | Structured interview | 4.3% (2) |
| | Structured questionnaire | 14.9% (7) |
| | Unspecified interview | 10.6% (5) |
| | Unspecified questionnaire | 10.6% (5) |
| **Setting** | *Locality* | |
| | Rural | 40.4% (19) |
| | Urban | 55.4% (26) |
| | Peri-urban | 4.3% (2) |
| **Dominant Water Supply** | *Standard of water supply* | |
| | Improved | 59.6% (28) |
| | Unimproved | 40.4% (19) |
| | *Dominant water source* | |
| | Borehole/tube-well | 14.9% (8) |
| | Communal water kiosk | 2.1% (1) |
| | Groundwater | 6.4% (3) |
| | Open well | 2.1% (1) |
| | Piped water connection (in the dwelling, yard or from neighbour) | 27.7% (13) |
| | Public standpipe | 10.6% (5) |
| | Tanker | 2.1% (1) |
| | Unclassified well | 21.3% (10) |
| | Unprotected well (including shallow dug wells) | 10.6% (5) |
| | *Dominant water availability measure reported* | |
| | Quantifies the amount of water available/used in a given time | 89.4% (42) |
| | Hours of supply a day/week | 6.4% (3) |
| | Frequency of breakdowns within a supply system | 4.3% (2) |

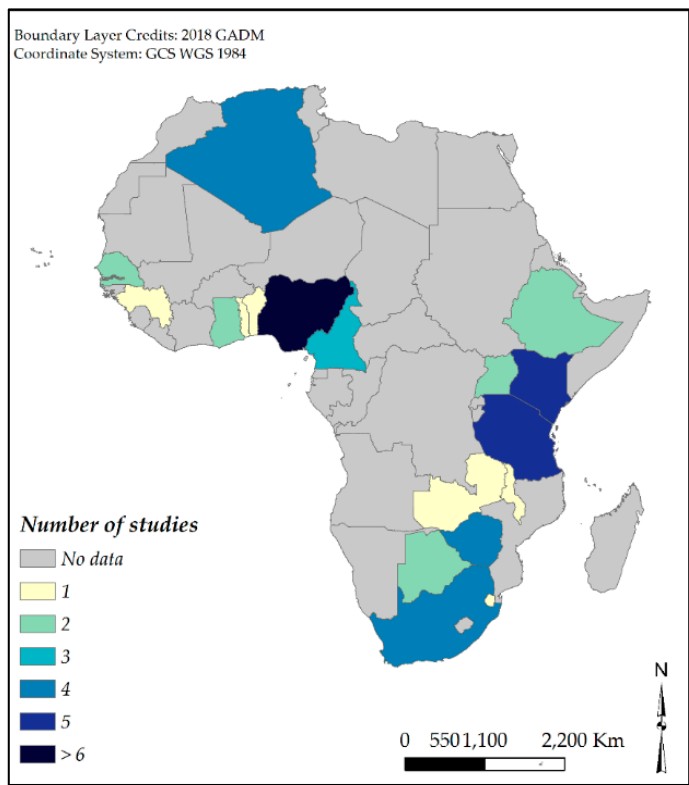

**Figure 2.** Geographical distribution of included studies.

*3.3. Monitoring Drinking Water Availability*

The majority of the studies (89%) reported water availability using either LPCD or LPHHD. The other two measures, hours of supply and the frequency of breakdowns, were reported by only 11% of studies. Hours of supply were reported per day [41] or per week [44], whereas the frequency of breakdowns were reported in an informal manner, often without quantification.

3.3.1. Measure One: Quantity of Water Available

Figure 3 shows per capita water availability based on locality, water supply status and the sampling strategy and size. Figure 4 shows the same but for household drinking water availability. Where studies have a sample size based on individuals, rather than households, the median household sample size has been used as a proxy.

The smallest sample sizes of less than forty households were in studies from Vogel et al. [45] and Hadjer et al. [43]. Small sample sizes were more common in studies researching rural areas. Studies which had large, representative samples included Hall and Vance [46], Tumwine et al. [40], and Adekalu et al. [38], with the latter sampling 20,000 households. Eleven urban studies used non-representative sampling strategies compared to eight rural studies. Studies that did not report their sampling strategy had comparatively small sample sizes below 500 households, e.g., [42].

In comparison, in Figure 4 where household drinking water availability is reported, the majority of sample sizes were less than 200 households; Juma et al. [47] had the smallest sample size of 98 households. Study site locality and whether it was urban or rural had minimal effect on sample size. Most studies used unrepresentative sampling strategies, with some exceptions (e.g., [48,49]).

Unimproved water supplies were more common in rural areas than urban areas (Figure 3). For example, 44% of studies reported using unimproved sources as their dominant water supply in urban areas, compared to 63% in rural areas. The greatest amount of drinking water available in an urban study was 130 LPCD [50], compared to 47 LPCD [51] in a rural study. Overall, the lowest reported water availability in rural areas was 4 LPCD [52] and 9 LPCD [42] in urban areas.

Overall, only four studies reported water availability which met the WHO benchmark of between 50 and 100 LPCD: all were urban study sites. Smith [50] reported the greatest amount with an average of 130 LPCD. All however used unrepresentative sampling strategies and small sample sizes. No study conducted in a rural location reported average daily water availability which met this WHO benchmark, although Kanda et al. [51] marginally missed the benchmark.

Figure 4 shows that of the studies which had urban study sites and improved water supplies, 80% had over 400 LPHHD. In contrast, Juma et al. [47] reported having higher LPHHD in a peri-urban study site than the three highest rural areas, with 340 LPHHD. Booysen [39] reported the highest amount with households in South Africa having 540 litres a day. Despite being an urban study site and the dominant supply being improved, Daramola and Olawuni [53] report considerably less available water at 205 LPHHD in Nigeria.

All rural studies reported having average water availability per household as less than 300 litres per day. Mijinyawa and Dlamini [54] report the lowest at 40 LPHHD, with an average household size of eleven people, thus suggesting that per capita water availability is likely to be significantly lower than the WHO benchmark. Only three studies [37,46,55] reported improved water supply use in rural areas. In all three, average drinking water availability was less than 100 LPHHD.

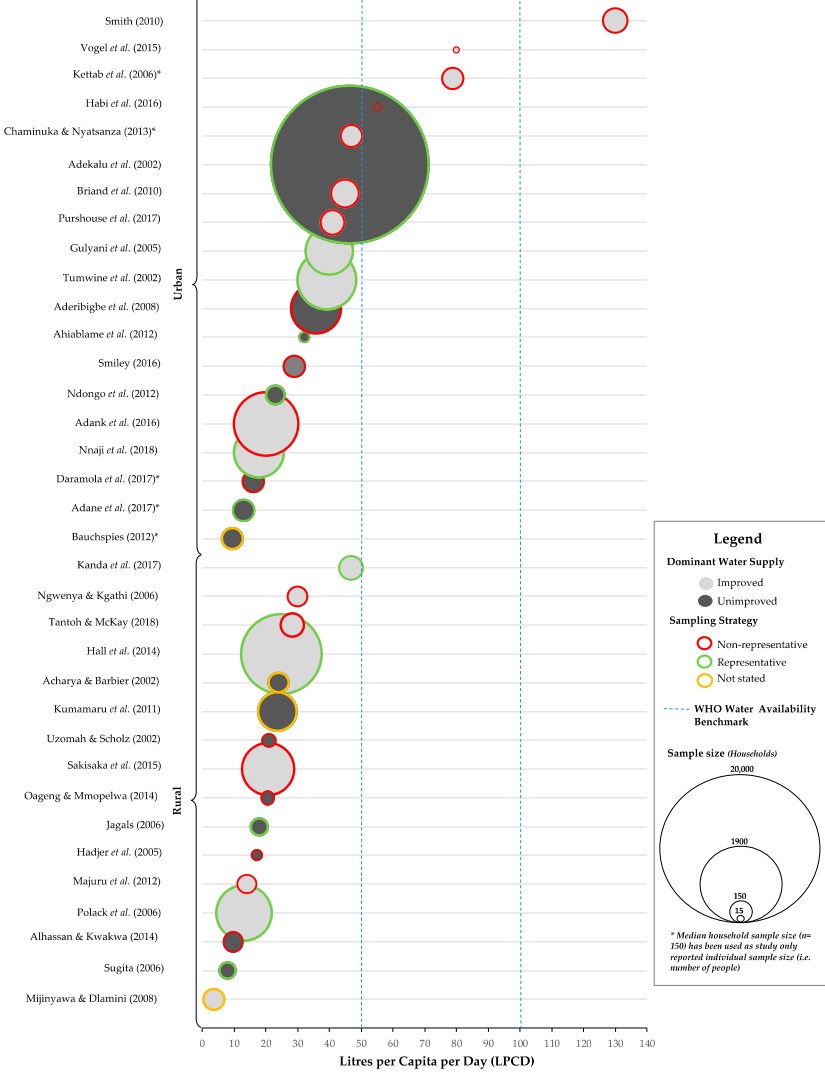

**Figure 3.** Daily per capita drinking water availability based on dominant household supplies, relative to the World Health Organization's (WHO's) benchmark of 50 to 100 LPCD [27,36,38,40,42,43,45,46,50,51,54–78].

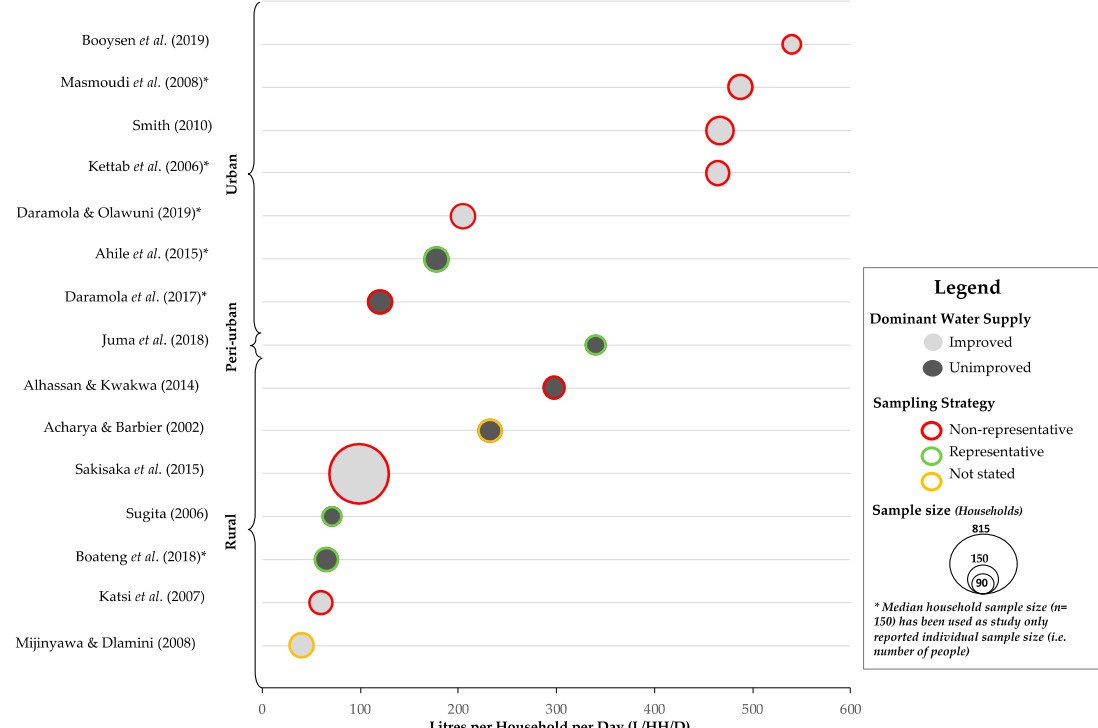

**Figure 4.** Daily household drinking water availability, based on dominant household supplies [27,37,47,49,50,54–56,67,71,78,79].

### 3.3.2. Measure Two: Hours of Supply

Three studies [41,44,52] reported hours of supply within a given time period. A further two studies [61,62] reported hours of supply as a secondary availability measure.

Intermittency in drinking water supply ranged from a supply being unavailable for less than two hours a day, to disruptions lasting a whole day or more. Oumar and Tewari [41] note that 64% of households had water available for only two hours a day, whereas Adams and Smiley [44] reported that 77% of their study sample lacked a continuous supply. More often than not, discontinuous supplies were piped water connections or public standpipes. For example, Gulyani et al.'s [61] respondents report that '36% of the households with private connections, [ . . . ] and 47% of those with yard taps had water available for less than 8 per day [61].

Common categories used in reference to days of supply a week included once every two weeks, once a week, one to two days, three to six days or the optimum of seven days. Aderibigbe et al. [62] found the majority of their respondents (53%) had a continuously available water supply, whereas Adams and Smiley [44] reported only 40% of households had water available seven days a week. However, in contrast, Adams and Smiley [44] reported that between 25% and 30% of their study sample had water available once every two weeks. Rugemalila and Gibbs highlight issues with piped connections, commenting that ' . . . even those who are connected experience inadequate and unreliable water supply: 98% reported not receiving water every day' [52].

### 3.3.3. Measure Three: Frequency of Breakdowns

Only two studies formally reported supply breakdown frequency [80,81]. Metrics used to report the frequency of breakdowns of a water supply were highly varied and primarily involved discursive prose or comment. For example, Machingambi and Manzungu noted the frequency of breakdowns of multiple supply types '12.1% of boreholes broke down very often, as did 6% of canals and taps; 5% of deep wells broke down an average amount, as did 3% of dams' [81]. In contrast, Kulinkina et al.'s reporting of breakdowns involved discussion of their respondents experiences,

for instance, a respondent using a piped water supply remarked that ' . . . sometimes water does not flow from the system for up to three days, even though there is no power outage and residents are not given a reason' [80], (p. 297).

*3.4. Modelling Drinking Water Availability*

An initial multiple linear regression model ($n$ = 34) (Appendix F, Table A4) was run and standard regression assumptions were checked. One outlier [50] was identified and subsequently removed from the dataset, before both the exploratory bivariate models and final multiple linear regression model were re-estimated.

Exploratory bivariate regressions (Appendix G, Table A5) identified significant relationships between LPCD with urban study location ($F(1,31)$ = 8.53, $p$ = 0.0065) and water scarcity ($F(1,31)$ = 4.64, $p$ = 0.039). The model containing urban location explained more of the variance in LPCD ($R^2$ = 0.22, adj.$R^2$ = 0.19), than the model with water scarcity ($R^2$ = 0.13, adj.$R^2$ = 0.10). Water scarcity was inversely related to LPCD in this model.

Results from the final multiple linear regression model (Table 5) show that the independent variables explained 61.4% of the variance in LPCD ($R^2$ = 0.61, adj.$R^2$ = 0.53). Of the five explanatory variables, only two (GDP per capita (PPP) and locality) were significantly related to LPCD when controlling for all other independent variables, $p$ < 0.05. This, however, only holds in urban locations, where a USD1000 increase in GDP per capita (PPP) increased water availability by 4.52 LPCD ($p$ = 0.001). In rural localities, a relationship was not found between GDP per capita (PPP) and LPCD. Evidence of a relationship between LPCD and piped supplies was present, however this was not significant ($p$ = 0.138). Water scarcity ($p$ = 0.608) and an intervention study design ($p$ = 0.924) were also insignificant predictors of LPCD.

**Table 5.** Multiple linear regression model of reported litres/capita/day (LPCD) from dominant household supplies across Africa, based on a subset of studies published between 2000 and 2019.

| Final Model | |
|---|---|
| GDP (per capita) (USD)/1000 | 0.055 (0.68) |
| Water Scarcity | 0.766 (1.47) |
| Urban study | −6.158 (7.58) |
| Intervention study | 0.526 (5.50)) |
| Piped water connection (in yard/house) | 9.399 (6.14) |
| Urban *GDP (per capita) | 4.461 (1.25) ** |
| Constant | 17.865 (6.15) * |
| Observations (n) | 33 |
| R-squared | 0.614 |

Note: Estimated standard errors in brackets. * significant at 5% ** significant at 1%.

## 4. Discussion

In almost all the studies included in this review, the WHO international benchmark for the amount of water available to households was not met. Except for Smith [50], Vogel et al. [45], Habi et al. [57] and Kettab et al. [56], the mean reported water availability was less than 50 LPCD (Figure 3). This pattern emulates findings reported by the International Benchmarking Network (IBNET) for Water and Sanitation Utilities. Figures from water providers and utilities across Africa show that water availability in five out of thirteen countries, and 81 of 173 municipalities falls below the WHO benchmark [82]. While this offers a more optimistic picture than findings from this review, it is evident that a significant proportion of urban water supply systems provide less than 50 LPCD. Nevertheless, Tamason et al. [2] provide further systematic evidence, including the rural perspective, to suggest the 50–100 LPCD benchmark is rarely being met. Settings that are reliant on non-piped water and in rural localities remain under-studied. However, in examining water availability in rural versus urban locations, our review responds to calls [15] for research which includes the rural perspective.

Results suggest lower domestic water availability in rural than urban areas, thus supporting known rural-urban differences in access to improved sources and water safety [83]. We also find, however, that this disparity between localities in the quantity of water available is not as significant as one could arguably expect. This suggests that while developing improved supplies in rural areas and rapidly developing peri-urban areas is critical, urban areas also require a focus to ensure water is available continuously. This is especially the case considering that classifications of rural, peri-urban and urban areas over-simplify a complex continuum [84].

Our quantitative analysis further explores the relationship between locality and water availability, by finding that GDP per capita (PPP) significantly increases reported LPCD in urban areas only (Table 5). This provides evidence to suggest that greater levels of development increase the availability of drinking water in these urban locations. Conversely, this relationship was not present in rural areas where economic investments differ based on their scale, demand, institutions and finance [85]. Improvements to rural water supplies often encounter different challenges, with the ability to improve supplies through installing a piped network affected by both the physical artefact and the associated institutional construct [*ibid.*]. Although evidence is supplied to suggest that having a piped supply can increase availability, it is not conclusive. Thus, ensuring that economic development, especially the development of piped supplies, in rural areas is possible, is crucial in improving water availability.

Overall, the literature from this review provides an understanding of water supply interruptions and availability that previously has been lacking. In providing evidence from a range of countries and localities across Africa, it is evident that despite improved water sources being the dominant supply type, and piped supplies in particular having a significant impact on the quantities of water available, they seldom provide the quantities required to maintain a good standard of living, with some evidence that supply is not continuous. Evidence is provided to suggest that the prevalence of interruptions in a water supply is not dependent on whether the supply is improved or unimproved, rather that each supply type is prone to discontinuities. This review also shows that households across Africa continue to rely on unimproved water supplies, which by nature deliver poor quality water and subsequently compromise health [16].

Most included studies reported on the quantities of water available (LPCD), but few reported on the hours of supply or frequency of breakdowns. The JMP has recently published a core and expanded set of questions for household surveys that reflect the Human Right to Water [14]. The core question set, which is being incorporated into household surveys, includes a subjective assessment by the household on whether they had enough water in the last week. It is a binary variable, which does not discriminate between differing degrees of unavailability, nor does it detail specifics such as whether households are using stored water or water from multiple sources to meet their needs. It also sets a benchmark that allows for water services that are not continuously available to households. Nearly one billion people globally do not have water that is continuously available [1], therefore, deciphering the degree to which water is available is important. In order to address this issue, household surveys may wish to consider a stricter definition of availability or additional questions that cover other types of unavailability, to gain a more granular understanding of the challenges facing households.

Several excluded studies (e.g., [86,87]) quantified availability using methods such as sustainability frameworks and the Household Water Insecurity Access Scale (HWIAS). The latter involves the use of household containers and measuring tools such as jerry cans and plastic bottles of standard volumes. Similarly, the Household Water Insecurity Experience (HWISE) scale was specifically designed to produce comparable measures of availability, especially across disparate cultural, ecological and environmental settings [88]. The capacity to consider environmental elements is especially important given it allows for local water scarcity to be considered. While our results suggest environmental factors such as water scarcity are not associated with reduced LPCD, this is unexpected and likely a result of our analysis being reliant on a small sample size, with many studies using unrepresentative data. This lack of a relationship also supports our argument that current methods being used to quantify availability are insufficient, given that they do not account for local factors such as water

stress. Arguably, both these methods, the HWIAS and HWISE scale, could provide Supplementary Materials within household surveys to distinguish whether a source is available as per the JMP's core question. Although few studies in this review reported on breakdowns and supply interruptions in Africa, it is encouraging that they feature in the JMPs expanded question set and this may support uptake in future national household surveys and research.

It is evident from the literature reviewed that there is a current reliance on using LPCD/LPHHD as the main method of measuring water availability, supporting Tamason et al.'s [2] call for it to be a dominant method. For households however, when asked as part of a survey to quantify their daily water availability, using LPCD/LPHHD is not the most intuitive method. As this review finds, the use of technologies such as water meters to measure water availability is evident and wider use could address the problem of recalling the availability of water, though they must be used tentatively given the nature of intermittent supplies [89]. While water meters provide an objective measure of water consumption [2], they have typically been deployed on piped water supplies, which, as shown, are not widely available to all populations [4,90]. In the context of Africa and other developing regions, complicating factors, such as household use of multiple supply types for multiple uses [91], make the comprehensive quantification of water consumption even more challenging. Alternative approaches, such as the HWIAS and HWISE, could therefore offer resolutions to this issue.

As shown in this review, the realities of water use are complex, and a range of different supply methods are relied upon by different populations. Given this, the difficulties and complexities of measuring availability further reflect the inadequacies of using measures such as LPCD/LPHHD, which are not necessarily intuitive to household respondents, despite suggestions [2] that it should be the 'end goal' for measuring water availability. As such, using methods such as water meters is of particular interest in quantifying poor water availability through robust measurement. As Tamason et al. [2] note, developing methods that can cross-validate water use values are critical in ensuring precise measurements.

*4.1. Methodological Problems Affecting Included Studies and Review Findings*

The overall representativeness of this assessment of drinking water availability is largely hindered by the inherent methodological issues evident in the included studies. These in part are due to the diversity of methods used.

The majority failed to report household sizes or missing data, further complicating the representativeness of this study's findings. This is especially evident in the similarities of the quantity of water available in rural and urban locations; a relationship which we would have expected to have differed greatly [83]. Several instances of arithmetic error were identified when reviewing studies; thus, the correctness of some reported results must be considered. Despite random sampling strategies stated as being used, many studies either did not report their sampling strategy or used unrepresentative and non-random sampling approaches, potentially introducing selection bias [92]. Selection bias could have resulted from studies targeting areas of high water stress, undermining their representativeness.

The failure of most included studies to report any measures of uncertainty concerning their estimates of water availability prevented any meta-analysis from being undertaken. Without such measures of uncertainty, heterogeneity in study findings could only be explored graphically, by visualising study characteristics such as sample size and sampling strategy.

The regression performed used all the data that was possible, with one outlier study removed as it was a highly influential point. The results must be interpreted based on the data that is used: many of the figures are taken from unrepresentative small-scale studies. There does not appear to be a systematic bias of representative studies being more likely in urban or rural areas, but the estimates may be influenced by small scale, unrepresentative studies.

Measuring water availability is complex. In the African context, and that of most lower income countries, it is routine to use multiple sources in order to build resilience to water shortages [91]. The exact wording of questions and responses used by included studies to ask households about

water availability was very seldom available in published reports, nor were recall periods reported. It is therefore unknown what essential information was collected alongside details of drinking water availability, for example, the coverage of storage tanks and the nature of alternative supplies. Nearly half of included studies did not document the identity of the respondent. For those that did, this was often the household head (with the definition of this undefined), rather than the household member responsible for water collection and management, increasing the risk of recall bias.

Furthermore, it is difficult to know whether the quantities of water reported consider household use of multiple water sources. Few studies explicitly asked about the use of secondary sources, nor was it clear whether reported amounts of water consumed represented only one dominant supply. If the reported amount of water available does only represent one dominant supply, then the apparently low pattern of household water availability in our review could have been because of under-estimation and failure to account for multiple source use in many of the included studies.

A total of 87% of the studies (Table 4) were cross-sectional studies, which explored water usage and availability over a short period of time, from one day to two weeks. They did not consider the role of longer-term causes of intermittent water supplies, such as seasonality. The three longitudinal studies included in the review focus solely on a specific water supply intervention and its effect on availability. Given the nature of the included studies, we were thus unable to assess seasonality of intermittent supplies.

At review level, there is evidence of an anglophone bias in part, as a result of search terms being predominantly in English. This is despite studies reporting in English, French and Portuguese falling within the review's scope. The culture of publication and differences by language and location are likely to have further affected this. Other than the three studies from Algeria, the included studies were written in English, and most were conducted in Anglophone countries (Figure 2). Likewise, no data was available for fragile states, such as the Democratic Republic of Congo or Sudan, likely to suffer poor water availability because of impaired service delivery. Finally, by limiting the review's scope to only peer reviewed articles, potentially relevant grey literature has not been included.

*4.2. Future Research*

The JMPs core and expanded question set concerning water availability for international monitoring, is necessarily restricted in scope by the logistical and financial constraints of conducting national household surveys. Therefore, smaller scale studies of water availability will continue to be valuable going forward. This would enable related concepts, such as water demand and its implications for water availability, to be considered in conjunction. When reporting findings from such studies, we recommend reporting of measures of uncertainty around means and/or reporting of standard deviations alongside means. Doing so would make it possible for future reviews to conduct meta-analysis.

For quantitative studies, reporting on sampling strategies and missing data, as well as using representative sampling strategies, would significantly improve the usefulness of findings in informing policy. Alongside recall periods and the identity of household respondents (as per Tamason et al.'s [2] systematic review), it is unclear whether the current literature captures multiple source use through probing about secondary sources. All of these aspects need to be reported and advocated as common practice in water sector research. Research could also be undertaken to examine the sensitivity of current reported water availability to such study design aspects.

This review found that a specific subset of studies used water meters as their main method of measuring water availability. These studies were conducted in wealthier urban areas of richer nations, where piped water supplies are more prevalent [93]. There is, therefore, the need for water meters as a methodology to be used in a range of settings. It would also be appropriate to adopt the use of methods such as the HWISE [94] and HWIAS [86], where resources allow (such as in research studies), in order to provide more rigorous and nuanced estimates of availability.

Additionally, smaller scale, more detailed studies over a longer time period, using complex and/or multiple methods of measuring water availability (e.g., novel water meters; diaries; complicated multiple question methods like HWISE/HWIAS), alongside simple single questions that the JMP can reasonably include in their core/expanded question sets, could offer a valuable insight into the limitations of a single survey question as recommended by the JMP for the SDGs. This builds on Tamason et al. [2] through the need to cross-validate simple questions against more comprehensive and thorough methods. In undertaking more detailed studies, the full depth of the water experience and its different dimensions within a household or with a user, such as variations between weekdays and weekends, can also be captured. Finally, informal settlements and fragile states, which could have very specific water usage patterns, appear under-studied and therefore should be priorities for future study.

## 5. Conclusions

Despite the MDG target concerning safe drinking water being met in 2015, evidence from 43 studies suggests insufficient drinking water remains a substantial issue across Africa. Almost all studies reported average water availability below the benchmark of 50–100 LPCD. Water insufficiency was more severe in studies of rural localities than in urban settings. These findings are broadly consistent with utility-provided figures and earlier related systematic review evidence [2]. This review also responds to calls for research that is nuanced and provides a regional exploration of drinking water availability.

There is considerable diversity in study methods, primarily a reliance on household surveys with small sample sizes and a lack of detail in reporting, limiting the ability to draw quantitative patterns and undertake substantive meta-analysis. It is recommended that future studies report methods of measurement in order that variation in availability can be examined through systematic review. The findings of this review point to a significant gap in the evidence base concerning water availability, thus, there is an ongoing need for rigorous studies to understand how best to measure water availability. This is particularly the case given challenges such as widespread household use of multiple water sources.

**Supplementary Materials:** The following are available online at http://www.mdpi.com/2073-4441/12/9/2603/s1, Sheet S1: Characterisation of Studies. Sheet S2: Other availability measures.

**Author Contributions:** M.L.H.T., J.A.W. and A.A.C. designed and conceptualised the study with the guidance and contributions of R.E.S.B. M.L.H.T. undertook the formal analysis with the support of A.A.C., J.A.W. and M.N. who validated findings through subsequent analysis. Writing and preparation of the original draft was undertaken by M.L.H.T., with reviewing, editing and supervision undertaken by A.A.C., R.E.S.B. and J.A.W. All authors have read and agreed to the published version of the manuscript.

**Funding:** This research was supported by a PhD award from the UK Economic and Social Research Council (Grant No. ES/P000673/1). This funding body had no role in the design of the study.

**Conflicts of Interest:** The authors declare no conflict of interest.

## Appendix A

**Table A1.** The JMP's Expanded Questions on Water Availability.

| Element of Availability | Expanded Question |
|---|---|
| Availability of water supply | 'Is water always available from your main water source?' |
| Reason for unavailability | 'What was the (main) reason you were unable to access sufficient quantities of water when needed?' |
| Continuity of water supply | 'How many hours per day is water supplied on average?' |
| Discontinuity of water supply | 'In the past month, for how many days was water from this source unavailable when needed?' |
| Storage tanks | 'Does your household have a large storage tank?' If so, 'How many litres does the storage tank hold?' 'How many times has the storage tank been filled in the last week/month?' 'Has there been any time in the last week/month when you have not been able to store sufficient water to meet your needs?' |
| Storage vessels | 'Does your household store drinking water in small containers?' 'Can you show me?' *Observe whether containers are covered or not.* |
| Seasonal variations in availability | 'What is your main source of drinking water in the wet season and the dry season?' |

## Appendix B

Search Terms (Box A1) and Geographical Search Terms (Box A2).

**Box A1.** Search Terms.

---

**Searches using the chosen search terms followed this structure:**

[Continuity/interruptions/availability] AND [domestic water] AND [water supply type] AND [African country]

**The below search terms were used:**

(Availab* OR Reliab* OR Interrupt* OR Function* OR Predict* Or Shortage OR Break* OR Limit* OR Failure OR Efficient OR Effective OR Intermitten* OR Irregularit* OR Function* OR Continu* OR Ration* OR Disrupt* OR Restrict* OR "hours of service per" OR "hrs of service per" OR "days of service per" OR "supply hours" OR "supply hrs" OR "supply interruptions")

**AND**

((Water AND Drink*) OR "drinking water")

**AND**

(Household OR Homestead OR Domestic OR Neighbo*)

**AND**

(potable OR suppl* OR tap OR faucet OR pipe* OR utility OR reticulated OR standpipe OR spigot OR "distribution network" OR "household connection" OR "protected well" OR "unprotected well" OR hand* OR pump OR "rope pump" OR "dug well" OR bore* OR tubewell OR "tube well" OR "open well" OR "shallow well" OR "traditional well" OR "covered well" OR "lined well" OR "rehabilitated well" OR windlass OR "drilled well" OR "ring well" OR "artesian well" OR "hand drawn" OR "hand-drawn" OR pumped OR fountain OR cistern OR "water dispenser" OR "municipal water" OR tank* OR truck OR "sand-filtered" OR "treated system" OR vendor OR "water station" OR kiosk OR "ground water" OR "groundwater" OR "water source")

**All results were limited to publication years post-2000.**

---

**Box A2.** Geographical Search Terms.

("sub-Saharan Africa" OR "East* Africa" OR "West* Africa" OR "South* Africa" OR "North* Africa" OR "Southern African Development Community" OR SADC OR "East African Community" OR EAC)
**OR**
(Algeria OR Angola OR Benin OR Botswana OR "Burkina Faso" OR Burundi OR Cameroon OR Cameroun OR "Cape Verde" OR "Cabo Verde" OR Chad OR "Tchad" OR "Central African Republic" OR CAR OR "République centrafricaine" OR CAF OR Comoros OR "Côte d'Ivoire" OR "Ivory Coast" OR Djibouti OR DRC OR RDC OR "Democratic Republic of Congo" OR "Republic of the Congo" OR "Republic of Congo" OR Congo OR Egypt OR "Equatorial Guinea" OR "Guiné Equatorial" OR "Guinée équatoriale" OR "Guinea Ecuatorial" OR Eritrea OR Ethiopia OR Gabon OR "Gabonese Republic" OR "République gabonaise" OR Gambia OR Ghana OR Guinea OR Guinée OR "Guinea-Bissau" OR "Guiné-Bissau" OR Kenya OR Lesotho OR Liberia OR Libya OR Madagascar OR Malawi OR Mali OR Mauritania OR Mauritius OR Maurice OR Morocco OR Mozambique or Moçambique OR Namibia OR Niger OR Nigeria OR Rwanda OR Ruanda OR "Rwandese Republic" OR "Saharawi Arab Democratic Republic" OR "Sao Tome and Principe" OR "São Tomé e Príncipe" OR Senegal OR Sénégal OR Seychelles OR Sierra Leone OR Somalia OR "Somali Republic" OR "South Africa" OR "Suid Afrika" OR "South Sudan" OR Sudan OR Swaziland OR Eswatini OR Tanzania OR Togo OR "Togolese Republic" OR Tunisia OR "Tunisian Republic" OR Uganda OR Zambia OR Zimbabwe)

## Appendix C

**Table A2.** Study Quality Ranking and Criteria.

| | | Criterion | Explanation/Question | Calculated from Information Captured Elsewhere |
|---|---|---|---|---|
| Core criteria for *all* studies | (1) | Rationale for study stated | Do the authors describe the rationale for the investigation? | Y (where 'objective' is 'n/a') |
| | (2) | Rationale for chosen participants | Are the eligibility criteria (inclusion/exclusion) for study participants described? | |
| | (3) | Water supply characteristics documented | Water source/supply characteristics documented. | Y (where water source is 'n/a') |
| | (4) | Sampling strategy reported | Sampling strategy for study (e.g., purposive; simple random; multi-stage cluster, etc.) is described. For secondary studies; sources and dates of access are given (e.g., DHS). | |
| | (5) | Arithmetic error | Do all numbers/percentages reported in the descriptive statistics add up? | |
| | (6) | Limitations and bias | Limitations of the study are discussed, taking into account potential bias or imprecisions. | |
| Criteria for qualitative studies *only* | (7) | Participants recorded throughout | Have numbers of individuals at each of stage of the study been reported (e.g., those ineligible; declining to participate; unavailable for interview)? | |
| | (8) | Justification for methods | Has the choice of interviews, questionnaires, focus groups etc. been justified? | |
| | (9) | Replicability | Have methods been suitably described to enable their replication? | |
| | (10) | Extraction of data and themes | How have they reached the conclusions and are they justified? | |

**Table A2.** *Cont.*

| | Criterion | | Explanation/Question | Calculated from Information Captured Elsewhere |
|---|---|---|---|---|
| Criteria for quantitative studies *only* | (7) | Statistical methods | Have all statistical methods been described, to a sufficient level of detail whereby replication is possible? | |
| | (8) | Justification for variables | Has the use of all chosen quantitative variables been explained? | |
| | (9) | Missing data | How missing data was addressed has been clearly explained. | |
| | (10) | Precision | Are confidence intervals reported for continuity/water availability estimates? | Y (if you have a field for the confidence intervals/standard error of proportion) |

## Appendix D

Reasons for studies being excluded at the screening stage (Figure A1) and the characterisation stage (Figure A2).

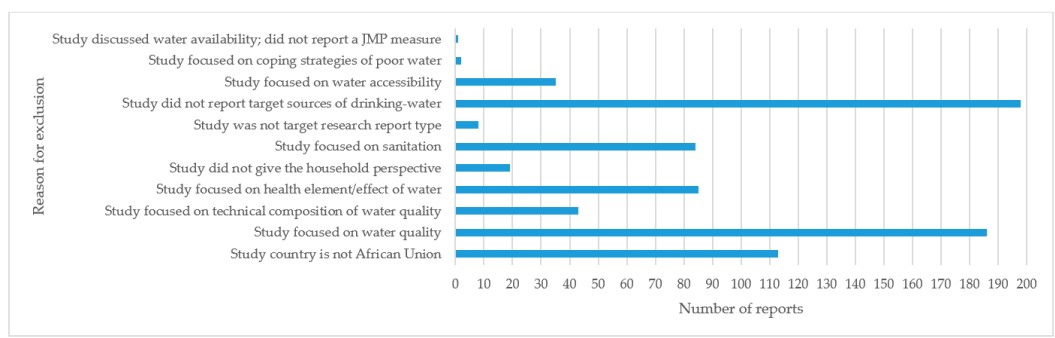

**Figure A1.** Reasons for reports being excluded at the screening stage. (NB: excluded from figure are reports which did not meet any inclusion criteria (*n* = 1490).

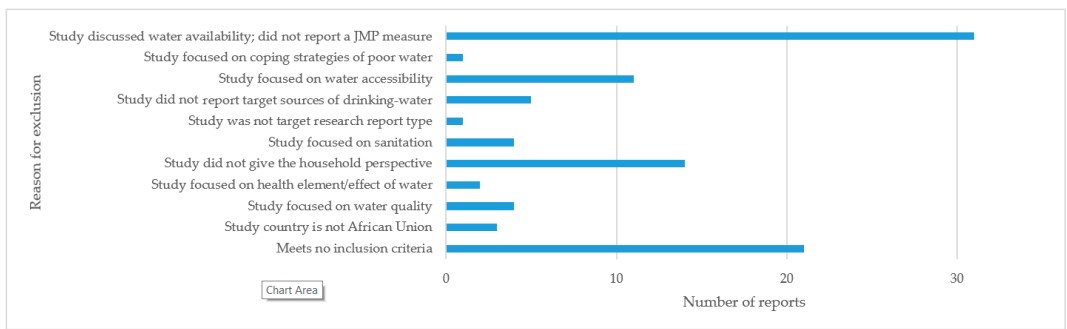

**Figure A2.** Reasons for studies being excluded at the characterisation stage.

# Appendix E

**Table A3.** Details of the sample sizes and their representativeness as reported by each published report.

| Sampling Details | | | Households | | | | Individuals | | | |
|---|---|---|---|---|---|---|---|---|---|---|
| Classification of Representativeness | Sampling Strategy Used by Studies | No. of Studies | Total Sample Size | Mean Sample Size | Range of Sample Sizes | No. of Studies | Total Sample Size | Mean Sample Size | Range of Sample Sizes |
| Representative | Simple random | 7 | 1685 | 241 | 40–761 | 6 | 3284 | 547 | 27–1080 |
| | Stratified-random | 2 | 2534 | 1267 | 674–1860 | 0 | - | - | - |
| | Systematic-random | 3 | 1183 | 394 | 50–1015 | 0 | - | - | - |
| | Systematic | 1 | 20,000 | - | - | 0 | - | - | - |
| Non-representative | Convenience | 2 | 200 | 100 | 60–140 | 2 | 693 | 347 | 40–653 |
| | Multi-stage | 10 | 4416 | 442 | 114–1203 | 2 | 1072 | 536 | 194–878 |
| | Purposive | 6 | 696 | 116 | 20–246 | 4 | 1201 | 306 | 22–683 |
| | Quota | 1 | 15 | - | - | 0 | - | - | - |
| | Self-selection | 1 | 103 | - | - | 0 | - | - | - |
| Unreported sampling strategy | | 3 | 727 | 242 | 130–447 | 1 | 230 | - | - |

## Appendix F

**Table A4.** Initial multiple linear regression models, including identified outlier [50] and excluding the outlier, of reported LPCD across Africa, based on a subset of studies published between 2000 and 2019.

| Variables | Initial Model with All Studies | Initial Model Excluding Outlier |
|---|---|---|
| GDP (per capita) (USD)/1000 | 2.399 (3.02) ** | 1.485 (2.26) * |
| Water Scarcity | 0.914 (0.43) | 2.394 (1.43) |
| Urban study | 17.328 (2.36) * | 14.750 (2.55) * |
| Intervention study | −6.672 (−0.81) | −2.336 (−0.36) |
| Piped water connection (in yard/house) | 20.482 (2.31) * | 9.491 (1.29) |
| Constant | 5.037 (0.61) | 7.511 (1.16) |
| Observations (n) | 34 | 33 |
| R-squared | 0.489 | 0.424 |

Note: Estimated standard errors in brackets. * significant at 5% ** significant at 1%

## Appendix G

**Table A5.** Exploratory bivariate regression models, of reported LPCD and each explanatory variable, based on a subset of studies ($n = 33$) published between 2000 and 2019.

| Variable | Coefficient | R-Squared |
|---|---|---|
| GDP (per capita) (USD)/1000 | 1.079 (1.41) | 0.061 |
| Water scarcity | 3.905 (2.16) * | 0.131 |
| Urban study | 16.812 (2.92) ** | 0.216 |
| Intervention study | −6.319 (−0.84) | 0.022 |
| Piped water connection (in yard/house) | 14.82 (1.86) | 0.100 |

Note: Estimated standard errors in brackets. * significant at 5% ** significant at 1%

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
