# Peer review of "Household-Reported Availability of Drinking Water in Africa: A Systematic Review"

_water, doi:10.3390/w12092603_

Round 1

Reviewer 1 Report

The paper "Household-reported availability of drinking-water in Africa: a systematic review" present the results of systematic literature survey related to issues on African domestic water availability. The literature survey includes the period form 2000 to 2019 and data available in five databases (Web of Science Core
23 Collection, Scopus, GEOBASE, Compendex and PubMed/Medline).

This paper is appropriately design, used methods are adequately described and obtained reuslts are clearly presented.

The English language and style are fine.

Author Response

We thank the reviewer for their positive feedback on the manuscript.

Reviewer 2 Report

Keywords: I suggest you add “intermittency “.

Global comments on the paper:

The paper has been improved since my first reviewing, for instance by including the water stress, the intermittency and several key issues. Many thanks to the authors for that improvement. Meanwhile I have some remaining comments and some new comments on the paper. You’ll find here-attached my main comments (I refer to my first reviewing comments number for easiness (I was the former reviewer number 3 of the first reviewing) for comments number lower or equal to 22):

  • Comment 13: Regarding water from neighbor, you indicate that it was not the dominant water source used in any of the included studies, nor was it one of the source types that was the focus of this review. Practically, I think you are wrong. I have been working for 30 years in African water supply projects. I confirm that in some cases, water from neighbor can have a strong impact. You must at minimum refer to that as a potential issue as it was not covered by the studies.
  • Comment 17: Regarding the seasonality of the intermittency and the seasonality of the poll, you should at least indicate that it could strongly impact the results and assessment of the results.
  • Comment 19: Regarding informal settlements, maybe you should mention that they have not been studied whereas they could have very specific results and should take attention of future studies.
  • Comment 24: Lines 127-129: So, you have considered these specific water supply when there were grouped with potable water network use. Is that correct?
  • Comment 25: Lines 131-135: Did you separate the week from the week end?
  • Comment 26: Lines 136-140: Could you argue why you avoid at least considering the proportion of total consumption linked with domestic uses? Indeed, in some case the other consumption could be very impactful, and in other cases they are secondary.
  • Comment 27: Lines 174-180. Don't you think this dominant water source could evolve with season? In such a case, how did you classify and assess that uses?
  • Comment 28: Lines 182-184: When considering that uncertainty, did you reassess the indicated uncertainty in order to avoid paper's inconsistency that sometime occurs.
  • Comment 29: Line 194: What do you mean by "its influence on review findings was considered". Be so kind to explain how you have considered that.
  • Comment 30: Lines 200-205: Did you include a specific uncertainty linked with that personal calculation you performed?
  • Comment 31: Lines 33--337: Could you provide the range of values and the standard deviation?
  • Comment 32: Lines 337-338: Same question as above mentioned.
  • Comment 33: Lines 345-350: If 60% are improved supplies and 21% unimproved or ambiguous, what corresponds to the remaining 19%? Be so kind to clarify.
  • Comment 34: Lines 378-380. Don't you have studies with mixed covered areas with urban and peri-urban areas?
  • Comment 35: Lines 403-405: It seems that it only corresponds to a part of the water used. Indeed, 4 LPCD and 9 LPCD seem to be very low and not in the order of level I get used in my own African projects. Don’t you think it suggests that there could be other water supply sources in these cases.
  • Comment 36: Lines 406-410: Don't you think it is also due to a focus of these studies on part of the cities where water stress occurred?
  • Comment 37: Lines 433-435: Practically you should also have intermittency with supply being available for more than 2 hours a day. No intermittency is 24h a day of water supply.
  • Comment 38: Lines 449-451: How these people can differentiate intermittent supply due to insufficient available water, from water supply breakdown?
  • Comment 39: Lines 542-544: This is effectively a central issue to look at.
  • Comment 40: Lines 552-555: the use of water meters with intermittent supply is an issue.
  • Comment 41: Lines 576-579: It is effectively a weakness of most of these studies. They lack a robust methodology.
  • Comment 42: Lines 594-597: Here again, it is a strong weakness of their methodology as it can lead to non-representative set of respondents.
  • Comment 43: Lines 598-602: It is true that under water stress, people tend to develop adaptative solutions, leading to several water supplies.
  • Comment 44: Global: There are lot of studies on these topics that are performed by engineering companies. Meanwhile, most of them are not published as it is not the target of the local authorities. But one of the limits of your approach is a lack of French speaking papers. There are for instance pHD thesis on these subjects.
  • Comment 45: Lines 604-609: As above mentioned there are French speaking studies, but they are not often published in international scientific journals.

Round 2

Reviewer 2 Report

All of my comments have been considered by the authors. Thanks for them.

This manuscript is a resubmission of an earlier submission. The following is a list of the peer review reports and author responses from that submission.

Round 1

Reviewer 1 Report

The manuscript “Household-reported availability of drinking-water in Africa: a systematic review” is very interesting. The work is well organized and comprehensively described. 

Reviewer 2 Report

The topic of this paper is very interesting and I really think it might be a very useful review for researchers and NGOs working in Africa.

GENERAL comments

However, I think the article explains too much about the method used to select the previous works and very few on analyzing them critically. The text is very repetitive in the low quality of the data and it mentions practically nothing about reasons or deep analysis extracted from the data. Moreover, I miss some theory on SDGs or at least, in sustainable development.

The main problem that I found in this paper, and because of that, I would not recommend its publication in this journal, is that it is not a originally research on the reasons of why SDG 6 is not accomplished in Africa. On the other hand, it is not also a quantitative meta analysis review of the data on this issue. I recommend the authors to decide if they want to rewrite the paper following one or the other option.

Reviewer 3 Report

Keywords: I suggest to add “intermittency “.

Global comments on the paper:

The paper is interesting as it presents a global assessment of publications on drinking water available for households in African countries, that is a key issue. Meanwhile the paper/methodology suffers from several key insufficiencies mainly due to the poor quality, homogeneity, and representativeness of the papers used for the review. You’ll find here-attached my main comments:

  • Comment 1: Global comment: English is fine.
  • Comment 2: Line 79: The questionnaire doesn’t’ include the required amount of water, the number of people in the household, the coverage of the storage tank. What about the nature of alternative water supply (distance, source, quality...)? What about potential sanitary issues? Do you consider that this essential information is never included in the questionnaires of the published papers you reviewed?
  • Comment 3 Line 122: It could have been useful to add the water demand if available
  • Comment 4: Line 179: “How much water do you need in a day?” could have been a good question to ask for.
  • Comment 5 Lines 196-201: What about the number of samples used for each study? The uncertainty of the results depends of the number of interviews, but also of the representativeness of the targeted samples. How did you evaluate this representativeness?
  • Comment 6: Line 224: Don't you think that the value should be 4961-2406=2585 rather than 2406?
  • Comment 7: The text is not clear enough. For instance, if you exclude 2264 from 2406 you should find 142 and not 240 (line 233)?
  • Comment 8: Lines 221-239: This paragraph is not totally clear, and I had to look at table 3 and more especially at figure 1 to understand. It means that figure 1 should be moved to the beginning of this paragraph and referred to.
  • Comment 9: table 4 Line 288: Globally the quality seems to be rather poor. How could we be sure of the representativeness of the assessment based from these studies?
  • Comment 10: Table 4 Line 288 item 7: This is a clear issue. Indeed, you indicate that only 17% of the samples meet the criteria. Moreover, I would personally have added the representativeness of the number (that could be an issue).
  • Comment 11: Line 289, table 5: How did you evaluate the representativeness of the samplings?
  • Comment 12: Line 289 Table 5: Don't you think it could have been useful to add the potential impact of contextual aspects (more especially regarding the hydrology and the habitat)?
  • Comment 13: Line 289 Table 5: Could you clarify some issues linked with that list: Didn't you have some dominant water source that could be in several sources (such as open unprotected well, or shallow dug well...). Your sources appear not necessarily separated from each other. What about water from neighborough?
  • Comment 14: Lines 308-309: I question the 316 household samples size compared to the 431 people samples size. What is the mean number of people per household. This key data is not provided.
  • Comment 15: Line 382: As I have indicated, these values are totally contextually dependent. You should for instance consider the local water stress.
  • Comment 16: Line 394: It should be very useful to have the number of people per household.
  • Comment 17: Lines 422-429: What about the seasonality of the intermittency?
  • Comment 18: Line 450: It would have been essential to look at potential reason of not fulfilment of MDG's goal. Unfortunately, this essential issue is not referred to. Could you explain if it was not assessed in the referred papers, or if you have forgotten to discuss it?
  • Comment 19: Lines 469-474: Do you have some papers targeting informal settlements, that have very specific water supply systems?
  • Comment 20: Lines 504-506: Take care of the fact that meters are generally not efficient in case of intermittent water supply.
  • Comment 21: Lines 539-541: This is effectively a key issue.
  • Comment 22: Lines 545-546: This effectively conducts to focus on English speaking countries, and water issues are not necessarily the same as in other African countries.
  • Comment 23: Line 548-550: I effectively suspect that reality is worse than what is described, as data are collected where they exist.

Reviewer 4 Report

The review paper "Household-reported availability of drinking-water in 2 Africa: a systematic review" present a a comprehensive and detailed overview of drinking-water availability across Africa.

The review examined drinking-water availability from the aspect of assess drinking-water availability across Africa from a household’s perspective and critical overwiev of used to measure drinking-water availability.

The authors detaily described advantages and disadvantages of each described study, as well as selection method for determination of relevant studies took in consideration.

I found this paper will written and interesting due to the presented results and described data elimination procedure and therefore, I recommend acceptance and publication of this review after minor spell check of English language and style.